# Lower probability and shorter duration of infections after COVID-19 vaccine correlate with anti-SARS-CoV-2 circulating IgGs

**Chiara Ronchini**[1]*, **Sara Gandini**[2], **Sebastiano Pasqualato**[2], **Luca Mazzarella**[2], **Federica Facciotti**[2], **Marina Mapelli**[2], **IEO Covid Team**[2,3¶], **Gianmaria Frige'**[1], **Rita Passerini**[3], **Luca Pase**[4], **Silvio Capizzi**[5], **Fabrizio Mastrilli**[5], **Roberto Orecchia**[6], **Gioacchino Natoli**[2], **Pier Giuseppe Pelicci**[2]*

1 Clinical Genomics, European Institute of Oncology IRCCS, Milan, Italy, 2 Department of Experimental Oncology, European Institute of Oncology IRCCS, Milan, Italy, 3 Division of Laboratory Medicine, European Institute of Oncology IRCCS, Milan, Italy, 4 Occupational Medicine, European Institute of Oncology IRCCS, Milan, Italy, 5 Medical Administration, CMO, IEO, European Institute of Oncology, IRCCS, Milan, Italy, 6 Scientific Directorate, European Institute of Oncology IRCCS, Milan, Italy

¶ Membership of the IEO Covid Team is provided in the Acknowledgments.
* chiara.ronchini@ieo.it (CR); piergiuseppe.pelicci@ieo.it (PGP)

**Data Availability Statement:** The complete dataset used for our analysis contains sensitive and potentially identifying information, considering that they relate to workers within our Institute. Data will

## Abstract

The correlation between immune responses and protection from SARS-CoV-2 infections and its duration remains unclear. We performed a sanitary surveillance at the European Institute of Oncology (IEO) in Milan over a 17 months period. Pre-vaccination, in 1,493 participants, we scored 266 infections (17.8%) and 8 possible reinfections (3%). Post-vaccination, we identified 30 infections in 2,029 vaccinated individuals (1.5%). We report that the probability of infection post-vaccination is i) significantly lower compared to natural infection, ii) associated with a significantly shorter median duration of infection than that of first infection and reinfection, iii) anticorrelated with circulating antibody levels.

## Introduction

SARS-CoV-2 pandemic has resulted in more than 220 million infections and 4.5 million deaths worldwide (Worldometer COVID-19 coronavirus pandemic. https://www.worldometers.info/coronavirus). SARS-CoV-2 vaccination induces strong humoral [1, 2] and cellular [3] immunity and its high effectiveness has been shown in different contexts and populations [4–9]. Knowing the duration and extent of the protection from SARS-CoV-2 infection in individuals who have recovered from COVID-19 or have received the SARS-CoV-2 vaccination is critical to determine the future dynamics of SARS-CoV-2 circulation and have direct impacts on non-pharmaceutical interventions, public health control measures and vaccination strategies. These pieces of information, however, are still an open issue.

## Study design

We performed systematic sanitary surveillance of the personnel working at the European Institute of Oncology (IEO), a large comprehensive cancer center in Milan, Northern Italy. Starting

be made available upon request to the corresponding authors and Dr. M. Monturano (massimo.monturano@ieo.it), head of the Health Risk Management and responsible for health data and security of IEO, following permission of release from the Institutional review board of IEO.

**Funding:** Financial support for our study was provided by Fondazione Europea Guido Venosta, National Instruments Corporation, Ralph Lauren and MFS Investment Management. Additional costs were covered through Institutional funds. The funders had no role in study design, data collection and analysis, decision to publish, or preparation of the manuscript.

**Competing interests:** The authors have declared that no competing interests exist.

from April 2020, all workers, including health-care, support staff, administrative and research personnel, were tested for SARS-CoV-2 infection by quantitative PCR (qPCR) detection of viral genes, using the Allplex SARS-CoV-2 Assay (Seegene) on nasopharyngeal or saliva samples. In order to compare the performance of saliva samples and nasopharyngeal swabs with our molecular assay for detection for SARS-CoV-2: i) we analyzed 9 saliva samples collected from symptomatic COVID-19 patients, positive for nasopharyngeal swab. All saliva samples (9/9, 100%) confirmed the positivity for SARS-CoV-2 (data not shown); ii) we collected and analyzed in parallel 47 saliva samples and nasopharyngeal swabs from individuals participating in our study. We obtained concordant results for 96% (45/47) of samples. Only 2 samples positive on nasopharyngeal swab for only the N viral gene with Ct cycles>37 scored negative on saliva (data not shown). All other positive cases gave highly comparable results, in term of Ct amplification, on both specimens. Based on these results and published data, which showed that saliva can be successfully employed for SARS-CoV-2 detection by molecular assays with similar or higher sensitivity compared to the same assays applied on nasopharyngeal swabs [10–13], we consider both specimens equivalent for our purposes and discuss them indistinctly throughout our manuscript. Humoral immunity was measured by testing levels of IgGs against the receptor binding domain (RBD) of the spike protein using an in-house ELISA assay [14]. Our assay showed high sensitivity (95.2%) and specificity (97.6%), that allowed monitoring IgG levels over time in healthy people as well as in COVID-19 patients with accuracy and reproducibility (see Materials and Methods for details and [14]). 1,493 participants were initially enrolled into the study starting from April 2020 and monitored before the vaccination campaign, which started on January 7th, 2021. 2,029 individuals, including the first cohort, were then vaccinated and monitored until June 2021 (characteristics of our study cohorts are reported in S1 Table of S1 File; timing of tests is described in Materials and Methods, 'Procedures' section, and S1 Fig in S1 File).

## Materials and methods

The institutional review board of the European Institute of Oncology approved the study (IEO 1271). Written informed consent was obtained from the participants.

### Study design and participants

SOS-COV2 is a prospective cohort study including staff working at the European Institute of Oncology in Milan, Italy. All health-care workers, support staff, and administrative staff working at hospital sites, who could provide written informed consent to participate in the study and anticipated remaining engaged in follow-up for 12 months, were eligible. Participants were excluded from this analysis if they did not participate to the screening after enrolment. Recruitment began in May 2020. Ethical approval was granted by the IEO ethical committee (IEO 1271).

### Statistical methods

We investigated the rate of infection/reinfection by positive status at baseline in the four groups identified by PCR and IgG (PCR- IgG-; PCR+ IgG+; PCR+ IgG-; PCR-IgG+, S2 Fig in S1 File). We collected information also on the values of Ct of genes for positive PCR and we did a further analysis including only reinfections with at least two positive genes. Participants reporting cough, fever, anosmia, or dysgeusia were defined as having COVID-19 symptoms.

We did univariate (Chi-square, Fisher exact tests and Wilcoxon rank tests) and multivariable logistic regression analyses to estimate Odds Ratios (ORs) to measure the association between the exposure (positive status at baseline) and infection/reinfection adjusting for significant confounders in order to identify independents factors associated with infections/reinfections.

Box-plots of IgG are presented by age, time and type of vaccine and curves of time to first infection/reinfection are presented and compared by Log-Rank tests.

## Procedures

At baseline, questionnaires on risk factors of exposures were sent electronically. SARS-CoV-2 antibody testing and real-time PCR (rtPCR) were performed at enrolment and at the end of the study. Furthermore, antibody testing was performed every 4 weeks. PCR test was performed after a positive serological test, in case of symptoms, after holidays and every 2 weeks for medical doctors. Swabs were taken by a trained professional (including anterior nasal swabs or combined nose and oropharyngeal swabs). COVID-19 vaccination was introduced into this cohort in January, 2021.

Participants were assigned to the positive cohort if they met one of the following criteria: antibody positive on enrolment or a positive PCR result at enrolment. Participants were assigned to the negative cohort if they had a negative antibody test and no documented previous positive PCR or antibody test.

A possible reinfection was defined as a participant with two positive PCR samples with a negative PCR between the two positive PCR samples and considering a positive PCR after 60 or more days, based on previous studies [15–18]. For this analysis participants with recurrent positive PCR results less than 60 days apart were not considered possible reinfections.

Data were collected on potential confounders, including profession and participant demographics, to permit adjustment in analysis.

The cohort susceptible to primary infection (PCR- IgG-): from first antibody-negative date to first positive PCR date or seroconversion (if no positive PCR test had been reported before seroconversion); or if neither of these occurred, to censor date. The cohort with previous infection (PCR+ IgG+; PCR+ IgG-; PCR-IgG+): the earliest date for previous infection was taken as whichever was first of the positive PCR result or the first positive antibody test (IgG>0.28).

The primary outcome was a reinfection in the positive cohort or a primary infection in the negative cohort, determined by PCR tests.

## SARS-COV-2 detection in respiratory specimens

Nasopharyngeal specimens were collected by trained healthcare professionals, while saliva samples were self-collected by the participants to the study, allowing at least one hour from eating, drinking and/or brushing of teeth before sample collection. Samples were stored at 4°C until use for processing, usually not more than 2 days after collection. Saliva samples were diluted 1:1 with Sputasol (per 100 ml: 0.1 g DTT, 0.78 g NaCl g, 0.02 g KCl, 0.112 g $Na_2HPO_4$, 0.02 g $KH_2PO_4$) and incubated for 5 min at room temperature, shaking at 500 rpm, in order to lose viscosity. For viral RNA extraction both Sputasol-treated saliva samples and nasopharyngeal swabs were inactivated with DNA/RNA shield (Zymo Research, Euroclone). Viral RNA was extracted from 300 ul of inactivated samples using the Sera-Xtracta Virus/Pathogen kit (Cytiva), following the manufacturer's instructions. Detection of the SARS-CoV-2 viral genes was performed by rtPCR using Allplex 2019-nCoV Assay and, more recently, the Allplex SARS-CoV-2 Assay from Seegene, following the manufacturer's specifications. Amplification of viral genes and data analysis was performed using the CFX96 Touch Real-time PCR Detection System (Biorad) and the Seegene Viewer platform, respectively.

## Serological tests for SARS-COV-2

Serological assays for SARS-CoV-2 were conducted as described [14]. Various commercial assays that utilize distinct viral antigens and detect different antibody classes are available.

However, SARS-CoV-2 serological tests available on the market do not always allow systematic simultaneous detection of a wide antibody spectrum for several antigens in a reliable and flexible manner. Conversely, serological enzyme-linked immunosorbent assays (ELISA) to detect immunoglobulins raised against the highly immunogenic receptor binding domain of the viral Spike Soluble Ectodomain (Spike) (RBD) provided robust results in terms of accuracy and reproducibility, that allow monitoring of IgG levels over time in healthy people pre- and post-vaccination, as well as in COVID-19 patients. Briefly, the recombinant Spike SARS-CoV-2 glycoprotein RBD was produced in mammalian HEK293T cells, purified by affinity chromatography, quantified and stored in liquid nitrogen. To detect immunoglobulins G (IgG) against the SARS-CoV-2 Spike RBD glycoprotein, purified RBD was adsorbed to a Nunc Maxisorp ELISA plate, aspecific binding was blocked by incubation with PBS-BSA 3% before applying patients' sera to be analyzed. Anti-RBD IgG presence was revealed with secondary anti-human-IgG antibody (BD, clone G18-145) conjugated to HRP by Glomax reading at 450 nm. The assay has been validated with a cohort of 56 COVID-19 subjects (severe, moderate and mild disease) and 463 (subjects collected in pre-COVID era, between 2012 and 2015). ROC curves have been implemented to determine the sensitivity and specificity of the assay, based on which IgG positivity was defined as absorbance at 450 nm >0.28 with a sensitivity of 95.2% and a specificity of 97.6% [14]. To work in the linearity range of the ELISA response, sera after vaccination were diluted either 1:200, 1:900 or 1:3645, and for the sake of clarity the OD at 450 nm was scaled to the 1:200 dilution before plotting.

## Results

### SARS-CoV-2 infections or re-infections prior to vaccination

In the pre-vaccination phase of our screening, we detected 266 SARS-CoV-2 infections (17.8%, 266/1,493). Multivariate logistic models were used to identify independent variables associated with infections during follow-up. Adjusting for age and symptoms, having a role as healthcare assistant in our Institute, specifically being a nurse or a physician vs. other professionals (researchers, technicians, administratives), was found to be highly correlated with increased probability of infection (S4 Table in S1 File, P<0.0001). Notably, subjects that were IgG+ at the time of enrollment (T0; S2 Fig in S1 File) had 66% significantly lower probability of having a positive swab (OR = 0.34, 95%CI: 0.14–0.80, P = 0.014, S4 Table in S1 File).

We also observed 8 putative re-infections (8/266; ~3%) (S2 Table in S1 File). Re-infections were defined as two PCR-positive samples interspersed with >1 PCR-negative samples. 5 individuals (all IgG+) had reinfection at >60 days. 7 of the 8 re-infected subjects were IgG+ at the time of enrollment (T0; S2 Fig in S1 File). Frequency of re-infection with respect to the status of IgG at time of enrollment was ~9% (7/80) in the IgG+ and 25% (1/4) in the IgG- subjects (difference is not statistically significant, Fisher exact test P = 0.335; Table 3). 6 (4/5 IgG+) showed rtPCR-positivity to only 1 of the 3 viral-genes tested and with Ct cycles >30. When considering only individuals testing positive for more than one SARS-CoV-2 gene in the PCR assay, frequencies of re-infection decreased significantly (2/266, <1%; 3% vs 0% for IgG+ vs IgG-).

### SARS-CoV-2 infections in vaccinated subjects

2,029 subjects were tested post-vaccination with the Pfizer BNT162b2 or Astra Zeneca (AZ) vaccines. 90% subjects completed the two doses of BNT162b2, and 181 received a single or double dose of AZ (Table 1). We observed a high rate of vaccination effectiveness, as measured by circulating anti-SARS-CoV-2 RBD IgGs one week post-vaccination, with: i) high antibody levels in the entire population (median ~5 fold increased over the threshold; min = 1 and

**Table 1. Study population.**

| Age group (y) | All Gender | Gender | | Tested post vaccination | | | Vaccine | |
|---|---|---|---|---|---|---|---|---|
| | Nr | M | F | Nr | M | F | BNT162b2 (1 dose) | AZD1222 (1 dose) |
| 19–29 | 456 | 145 | 311 | 429 | 138 | 291 | 354 (10) | 56 (12) |
| 30–39 | 510 | 193 | 317 | 483 | 185 | 298 | 424 (11) | 39 (14) |
| 40–49 | 547 | 161 | 386 | 531 | 158 | 373 | 488 (12) | 24 (10) |
| 50–59 | 451 | 155 | 296 | 432 | 153 | 279 | 406 (10) | 9 (12) |
| 60–69 | 132 | 56 | 76 | 130 | 55 | 75 | 122 (5) | 4 (0) |
| 70–81 | 25 | 17 | 8 | 24 | 16 | 8 | 24 (0) | 1 (0) |
| **Total Nr** | **2121** | **727** | **1394** | **2029** | **705** | **1324** | **1818 (48)** | **133 (48)** |

max = 12.5) and across each age-group (age range: 19-81y/o); and ii) only 1.9% (39/2,029) of non-responders (IgG levels <0.28) (Fig 1). IgG levels inversely correlated with age, with the lowest levels (median of 7.9) in subjects >70 (median of 20.0 in the age group 19–29; Fig 1). Moreover, levels of IgG monotonically declined over time post-vaccination, though 95.3% (1303/1367) or 98.4% (1030/1047) of tested individuals showed IgG levels above the threshold at 3 or 4 months post-vaccination, respectively (median of 2.22 and 1.57, respectively; Fig 1).

In the 2,029 vaccinated subjects, we identified 30 cases (1.5%) of molecularly-detectable infections (Table 2). 15/30 of these cases showed positivity for 2 or 3 viral genes out of 3 tested, while the remaining 15 were positive for only the N gene at Ct>35. However, 9 cases showed Ct ranges below 30 PCR cycles and one case below 20, suggesting efficient viral replication (Table 2). 4 had received only one dose of the AZ vaccine, while all others had completed the two doses of the BNT162b2. Notably, the probability of infection after vaccination was significantly lower than in the non-vaccinated subjects (1.47% vs 9.52%; P<0.0001; Table 3), confirming the effectiveness of vaccination [4–9]. Infections were detected in all age groups except

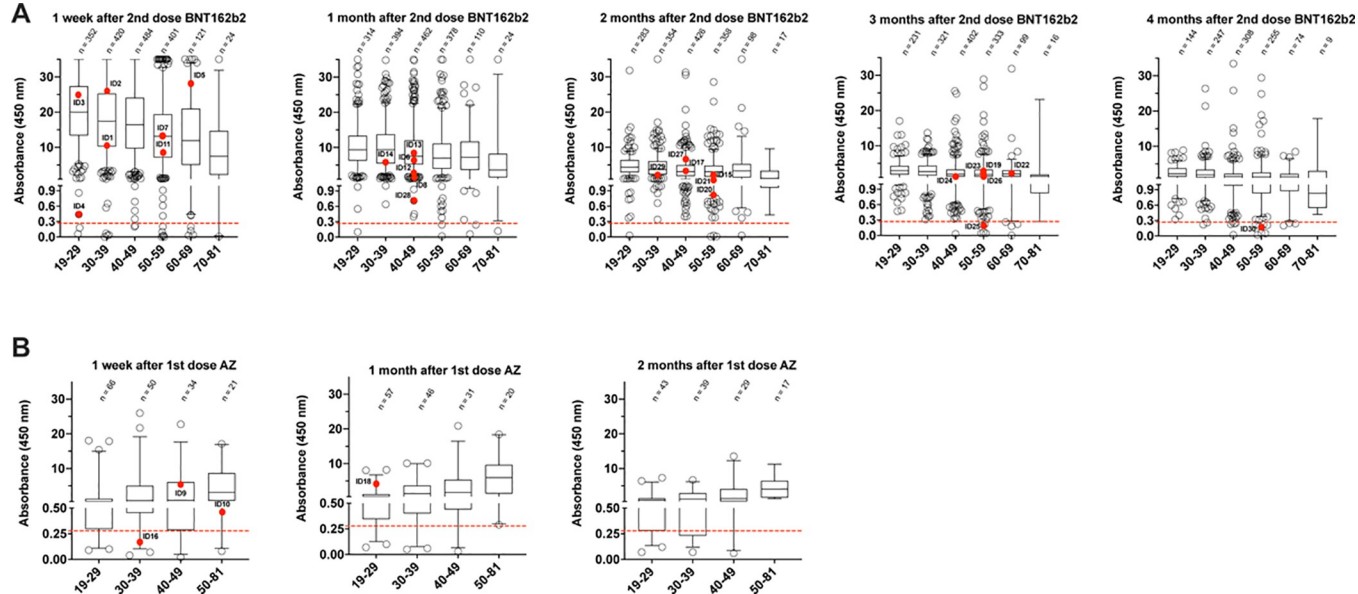

**Fig 1. IgG levels against the receptor binding domain (RBD) of the SARS-CoV-2 protein post-vaccination.** A, after 2 doses of BNT162b2 vaccine. Individuals are divided by age groups. The red dots highlight the IgG levels in individuals that resulted positive to SARS-CoV-2 infection by rtPCR. The dashed red line indicates the threshold of positivity for our serological test (positives>0.28). Boxes define the 25th and the 75th percentiles; horizontal line within the boxes indicates the median and whiskers define the 5th and the 95th percentiles. B, as for panel A after 1 dose of AstraZeneca (AZ) vaccine.

**Table 2. SARS-Cov2-positive individuals post-vaccination.**

| Subjects | Age range | Days post-vaccination | E gene | RdRP gene | N gene | anti-RBD IgG (range 0.28–35) | IgG quartile (min 1, max 4)* | Vaccine |
|---|---|---|---|---|---|---|---|---|
| ID1 | 36–40 | 7 | n | n | 38.96 | 10.65 | 1 | II jabs BNT162b2 |
| ID2 | 31–35 | 5 | n | n | 35.80 | 25.81 | 4 | II jabs BNT162b2 |
| ID3 | 26–30 | 7 | n | n | 38.32 | 24.67 | 3 | II jabs BNT162b2 |
| ID4 | 21–25 | 11 | 38.98 | n | 36.91 | 0.55 | 1 | II jabs BNT162b2 |
| ID5 | 61–65 | 8 | 31.41 | 35.04 | 33.14 | 27.66 | 4 | II jabs BNT162b2 |
| ID6 | 41–45 | 42 | n | 37.67 | n | 6.10 | 2 | II jabs BNT162b2 |
| ID7 | 45–50 | 12 | 30.29 | 33.73 | 31.92 | 12.11 | 2 | II jabs BNT162b2 |
| ID8 | 41–45 | 40 | 25.35 | 27.27 | 27.15 | 3.28 | 1 | II jabs BNT162b2 |
| ID9 | 41–45 | 23 | n | n | 38,9 | 5.84 | 3 | I jab AZ |
| ID10 | 51–55 | 21 | 22.85 | 25.45 | 24.56 | 0.45 | 2 | I jab AZ |
| ID11 | 56–60 | 46 | 21.76 | 23.78 | 22.76 | 9.48 | 2 | II jabs BNT162b2 |
| ID12 | 41–45 | 53 | 21.4 | 22.79 | 19.69 | 4.57 | 1 | II jabs BNT162b2 |
| ID13 | 46–50 | 55 | 36.58 | 38.78 | 34.4 | 8.64 | 3 | II jabs BNT162b2 |
| ID14 | 36–40 | 53 | 28.77 | 31.61 | 29.85 | 6.76 | 2 | II jabs BNT162b2 |
| ID15 | 51–55 | 67 | 29.06 | 31.89 | 28.65 | 1.21 | 1 | II jabs BNT162b2 |
| ID16 | 31–35 | 21 | 37.16 | n | 34.74 | 0.21 | 1 | I jab AZ |
| ID17 | 46–50 | 72 | n | n | 37.12 | 3.09 | 2 | II jabs BNT162b2 |
| ID18 | 26–30 | 55 | n | n | 37.34 | 4.98 | 4 | I jab AZ |
| ID19 | 51–55 | 98 | 21.05 | 23.17 | 20.56 | 3.25 | 3 | II jabs BNT162b2 |
| ID20 | 46–50 | 98 | 30.03 | 32.52 | 28.61 | 0.69 | 1 | II jabs BNT162b2 |
| ID21 | 51–55 | 91 | n | n | 37.01 | 1.59 | 2 | II jabs BNT162b2 |
| ID22 | 56–60 | 88 | n | 38.37 | n | 2.29 | 3 | II jabs BNT162b2 |
| ID23 | 46–50 | 68 | n | 38.83 | n | 2.88 | 1 | II jabs BNT162b2 |
| ID24 | 46–50 | 98 | n | n | 37.04 | 1.69 | 2 | II jabs BNT162b2 |
| ID25 | 51–55 | 99 | n | n | 36.88 | 0.26 | 1 | II jabs BNT162b2 |
| ID26 | 46–50 | 108 | n | n | 36.90 | 3.69 | 1 | II jabs BNT162b2 |
| ID27 | 41–45 | 75 | 35.89 | 36.94 | 35.87 | 6.50 | 4 | II jabs BNT162b2 |
| ID28 | 46–50 | 78 | n | n | 36.07 | 0.71 | 1 | II jabs BNT162b2 |
| ID29 | 31–35 | 116 | n | n | 37.25 | 3.06 | 3 | II jabs BNT162b2 |
| ID30 | 56–60 | 139 | 12.41 | 15.61 | 11.68 | 0.22 | 1 | II jabs BNT162b2 |

* quartiles normalized to age and time after vaccination; n, not detectable.

**Table 3. Frequency of natural reinfections and infections after vaccination by IgG status.**

| | | | | IgG- | | | IgG+ | | |
|---|---|---|---|---|---|---|---|---|---|
| | Natural infections pre-vaccine | Vaccinated Subjects | P-value | Natural infections pre-vaccine | Vaccinated Subjects | P-value | Natural infections pre-vaccine | Vaccinated Subjects | P-value |
| **All** | 84 (100%) | 2029 (100%) | | 4 (100%) | 53 (100%) | | 80 (100%) | 1967 (100%) | |
| **No PCR+ swab** | 76 (90.5%) | 1999 (98.5%) | <0.0001 | 3 (75%) | 50 (94.3%) | 0.259 | 73 (91%) | 1940 (98.6%) | 0.0002 |
| **PCR+ after natural infection and after vaccine** | 8 (9.5%) | 30 (1.5%) | | 1 (25%) | 3 (5.7%) | | 7 (9%) | 27 (1.4%) | |

IgG+ values are assessed at baseline for the cohort pre-vaccination and 1week post-vaccination for the second cohort.

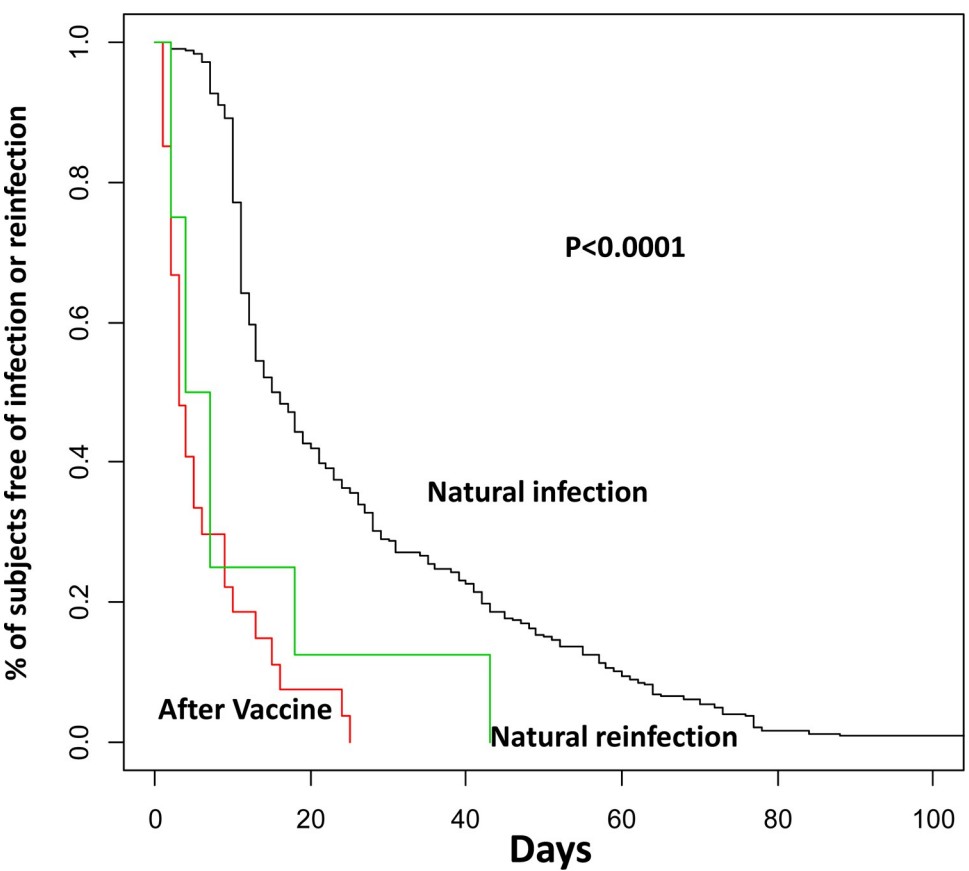

**Fig 2. Time of SARS-CoV-2 infections.** Kaplan-Meier curves of natural infections (black line); natural reinfections (green line); infections post-vaccine (red line). P-value, Log-rank test.

for the oldest (median 47.4 years old; min 23 and max 62; Table 2). Time of infection varied from few days post-vaccination to >4 months after completion of the vaccination protocol (min 5 days, max 139, median 55 days post-vaccination, Table 2).

The median duration of infections based on a positive PCR test in the vaccinated subjects was 2 days (Interquartile range—IQR: 2–4). Notably, this duration was significantly shorter than post-natural infections (16.5 days; IQR: 11–40.5; P<0.001) or re-infections (11 days; IQR 4–21; P = 0.0035) in the pre-vaccinated subjects, suggesting significantly shorter duration of viral shedding in vaccinated individuals as compared to the unvaccinated ones (S3 Table in S1 File and Fig 2). Moreover, to our knowledge, all infected individuals reported asymptomatic or pauci-symptomatic infections.

## Correlations with IgG levels of SARS-CoV-2 infections in vaccine responders

The frequency of molecularly detectable infections among the IgG+ vaccine-responders (subjects that positively responded to vaccination) was significantly lower (1.4%) than in the IgG + non-vaccinated subjects after natural infection (9%; P = 0.0002) and the IgG- vaccine non-responders (5.7%; P = 0.042) (Table 3).

Among the newly infected vaccinated-subjects, 3 cases were IgG non-responders (IgG-). Notably, the remaining 27 infected vaccinated individuals were mainly distributed in the lower quartiles of anti-RBD antibody titers (~74% in quartiles 1 and 2; Table 2 and Fig 1).

Moreover, very high antibody titers counter-correlated with the extent of viral replication, based on Ct amplification values of the viral genes (Table 2).

## Discussion

Our surveillance study yielded three main findings: i) the probability of infections after COVID-19 vaccine is lower than after natural infection; ii) the few SARS-CoV-2 infections occurring in individuals who mounted a serologically positive response to vaccination are of significantly shorter duration than the first infections in non-vaccinated individuals; iii) the levels of anti-SARS-CoV-2 circulating IgGs were inversely correlated with the frequency and duration of viral detection, as recently reported [19].

In our study we observed the occurrence of infection in vaccinated individuals with high viral titers suggestive of efficient viral replication. This is supported by previous studies that detected active viral replication in SARS-CoV-2-positive vaccinated individuals by analysis of subgenomic viral RNA [20, 21]. Nonetheless, similarly to our results, the frequency of infection post-vaccination was low, the symptomatology was really mild and, importantly, the viral load rapidly declined [20, 21]. In particular, in agreement with our data, in the one case report, the speed of viral decay was significantly faster compared to a reference group of non-vaccinated individuals [21]. Although vaccination is effective and protects from severe symptoms, these results suggest caution and the necessity of maintaining protective measures in order to avoid viral spreading even after vaccination.

Our cohort study in healthy workers conducted from the end of the first wave confirmed that reinfection after natural infection is seven times more likely than infection after vaccination. This finding supports the CDC recommendation that all eligible persons be offered COVID-19 vaccination, regardless of previous SARS-CoV-2 infection status. However, the probability of reinfection largely depends on pre-existent IgG positivity. Thus, serological testing in vaccinated individuals might help to identify the population at higher risk of infection.

Reinfections have been reported internationally since June 2020, although they remain uncommon: test results of 4 million people in Denmark found that < 1% of persons who tested positive for SARS-CoV-2 experienced reinfection [22]. The vastly shorter duration of post-vaccine infections likely has major impacts on models to predict epidemiological dynamics, which critically rely on this parameter [23, 24], and may suggest a modification of the isolation policies, which still recommend releasing from isolation 10 days after a first positive PCR test for asymptomatic testing, without distinction for vaccinated subjects [25].

The immune response to SARS-CoV-2 infection is highly complex and involves the interplay of both humoral and cellular components. In particular, B and T cell immune responses seems to be elicited in the majority of infected patients and to last for at least 6 months without showing decline (reviewed in [26]), in contrast to what we observed for IgG levels. Therefore, cellular mediated immunity could play a fundamental role in long-term response and protection from SARS-CoV-2 infection. However, these aspects are beyond the scope of our analysis, in which we aimed for the identification of inexpensive, rapid and reliable markers for the assessment of the risk of SARS-CoV-2 infection, especially in large cohorts and in environments frequented by fragile individuals, such as our Institute.

Large longitudinal cohort studies with regular testing are needed to provide systematic epidemiological, virological, immunological, and clinical data useful to understand the rates of reinfection and their implications for health policies. Moreover, the alfa variant started to spread in our country at the beginning of 2021 and became prevalent by the middle of March 2021. Therefore, during the pre-vaccination period of our study was not present in our country but became prevalent during the period of the vaccination campaign. Although we did not systematically address the issue of the viral strain infecting our cases, post-vaccination, all tested

cases were positive for the alfa variant. Considering that the delta variant was not diffuse in our country at the time of testing described in our study, the data presented will need to be updated to estimate the impact of the delta variant on reinfection/post-vaccine infection risk.

## Conclusions

Overall, our data show that individuals who responded to vaccination based on the detection of anti-RBD antibodies were still susceptible to SARS-CoV-2 productive infection, suggesting caution, especially for healthcare workers that are daily in contact with fragile patients, such as cancer patients in our Institute. However, the probability of infection after vaccination is rare and significantly less frequent compared to reinfection after natural infection, in particular in responders, which are the vast majority. Furthermore, the duration of infection in vaccinated individuals is significantly shorter to the one observed post-natural infection, suggesting that post-vaccination viral shedding is likely very limited, recommending for a revision of the isolation policies, that could drastically reduce the time of quarantine, with clear important social and economic implications.

## Supporting information

**S1 File.**
(DOCX)

## Acknowledgments

We are very grateful to Fondazione IEO-CCM for fund-raising.

Membership of IEO Covid Team (lead author and members, listed in alphabetical order):

Gioacchino Natoli[1]*, Federica Bellerba[1], Chiara Bozzetti[2], Stefania Brandini[1], Thelma Capra[1], Valentina Cecatiello[1], Giuseppe Ciossani[2], Errico D'Elia[1], Giuseppina De Feudis[1], Mirko Doni[1], Karin J. Ferrari[1], Laura Ferrari[1], Laura Furia[1], Barbara A. Gallo[1], Donatella Genovese[1], Elvira Gerbino[1], Stefania Guerini[2], Maria G. Jodice[1], Cinzia Massaro[2], Mariangela Massaro[2], Davide Merli[2], Claudia Miccolo[1], Erika Mileti[1], Manuela Moia[1], Francesca Montani[1], Silvia Monzani[1], Andrea Moro[2], Lorena Moretti[2], Adeline Ngounou Ngassa[2], Daniela Osti[1], Isabella Pallavicini[1], Rossana Piccioni[1], Elena Prosperini[1], Sara Raimondi[2], Cristina Richichi[1], Mauro Romanenghi[1], Adriana Salvaggio[1], Costanza Savino[1], Cristina Simone[2], Cristina Spinelli[1], Massimo Stendardo[1], Erika Tenderini[1], Cecilia Toscani[1], Gianluca Tripodi[1], Alessandro Verrecchia[1], Clara Visintin[1], Marika Zanotti[1]

[1]Department of Experimental Oncology, European Institute of Oncology IRCCS, Milan, Italy

[2]Division of Laboratory Medicine, European Institute of Oncology IRCCS, Milan, Italy

* *Lead author. Email*: *gioacchino.natoli@ieo.it*

## Author Contributions

**Conceptualization:** Luca Mazzarella, Roberto Orecchia, Gioacchino Natoli, Pier Giuseppe Pelicci.

**Data curation:** Chiara Ronchini, Sara Gandini, Sebastiano Pasqualato, Rita Passerini.

**Formal analysis:** Chiara Ronchini, Sara Gandini, Sebastiano Pasqualato.

**Project administration:** Luca Pase, Silvio Capizzi, Fabrizio Mastrilli.

**Resources:** Chiara Ronchini, Federica Facciotti, Marina Mapelli, Gianmaria Frige', Rita Passerini.

**Supervision:** Rita Passerini, Roberto Orecchia, Gioacchino Natoli, Pier Giuseppe Pelicci.

**Writing – original draft:** Chiara Ronchini, Sara Gandini.

**Writing – review & editing:** Chiara Ronchini, Sara Gandini, Luca Mazzarella, Federica Facciotti, Marina Mapelli, Gioacchino Natoli, Pier Giuseppe Pelicci.

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
