## [Decision Letter · Decision Letter 0]

4 Nov 2021

PONE-D-21-31133Lower probability and shorter duration of infections after Covid-19 vaccine correlate with anti-SARS-CoV-2 circulating IgGsPLOS ONE

Dear Dr. Ronchini,

Thank you for submitting your manuscript to PLOS ONE. After careful consideration, we feel that it has merit but does not fully meet PLOS ONE’s publication criteria as it currently stands. Therefore, we invite you to submit a revised version of the manuscript that addresses the points raised during the review process.

Please address all comments from reviewer 1.

In addition, please move relevant information from supplementary material (SM) to the body of the manuscript. The reader should be able to follow the study and the procedures used by reading the manuscript. Place in SM details and minor points that are not needed to understand the study. For example, information about the test for levels of IgG against virus receptor-binding domain is important and should be in the manuscript, not in the SM.

Description of statistical methods is also important, move it to the manuscript from SM. Only details that are not critical to understand the work should remain in SM.   

Table 1B shows data for three viral genes. These data are not explained or discussed in the manuscript. Why are these values in the table and what is their importance?

Line 71, sentence starting with "While age,.." is not clear. Specifically, what is "While age" referring to. Please revise.

We look forward to receiving your revised manuscript.

Kind regards,

Luisa Gregori, Ph.D

Academic Editor

PLOS ONE

Journal Requirements:

5. One of the noted authors is a group or consortium [IEO Covid Team]. In addition to naming the author group, please list the individual authors and affiliations within this group in the acknowledgments section of your manuscript. Please also indicate clearly a lead author for this group along with a contact email address.

Reviewers' comments:

Reviewer's Responses to Questions

**Comments to the Author**

1. Is the manuscript technically sound, and do the data support the conclusions?

Reviewer #1: Yes

2. Has the statistical analysis been performed appropriately and rigorously? 

Reviewer #1: Yes

3. Have the authors made all data underlying the findings in their manuscript fully available?

Reviewer #1: Yes

4. Is the manuscript presented in an intelligible fashion and written in standard English?

Reviewer #1: Yes

5. Review Comments to the Author

Reviewer #1: The study provides an interesting picture of the patients admitted to a single centre in 27 months. The study design is appropriated and the results substantially confirm the data of other studies concerning the vaccination against Sars-CoV-2. There are issues to be addressed:

• Concerning the sampling for the diagnostic analyses, how many enrolled subjects underwent naso-oropharyngeal swab and how many were sampled for saliva? If swab and saliva were obtained from the same subjects, did authors observed discrepancies related to the RT-PCR performed for samples correspondent to late stages of infection? Authors should clearly explain this information.

• During the study period different viral clades, including some VOCs such as the alpha variant, emerged and spread. Therefore, did authors observe differences in terms of viral load and serological measurements, among infected subjects in 2020 and those infected in the first part of 2021 (namely, those more probably exposed and infected with alpha variant)?

• Concerning the possible reinfections, there are three subjects who may be reinfected after less than 100 days. Authors could better discuss if the data reported in table S2 could be more indicative of viral remnants of the first infection. Could these values be non-specific? Instead, concerning those patients putatively reinfected after more days, may these patients be re-infected with a different Sars-CoV-2 variant? Authors should better discuss this section.

• The Discussion section could be improved. Indeed, authors should compare their findings concerning the vaccination with other data reported in literature (e.g. PMID: 34442817, 2021; PMID: 34249017, 2021). For instance, authors should mention the importance of cell-mediated immunity mediators as markers of effective response to vaccination with respect of antibody levels (e.g. PMID: 33534923, 2021). Moreover, authors could discuss if the stratification of the enrolled positive patients according to the viral strain could add major information.

6. PLOS authors have the option to publish the peer review history of their article (what does this mean?). If published, this will include your full peer review and any attached files.

Reviewer #1: No

---

## [Author Response · Author response to Decision Letter 0]

31 Dec 2021

Editorial comments

Please move relevant information from supplementary material (SM) to the body of the manuscript. The reader should be able to follow the study and the procedures used by reading the manuscript. Place in SM details and minor points that are not needed to understand the study. For example, information about the test for levels of IgG against virus receptor-binding domain is important and should be in the manuscript, not in the SM.

Description of statistical methods is also important, move it to the manuscript from SM. Only details that are not critical to understand the work should remain in SM. 

Response: As suggested, we moved all relevant information from the Supplementary Material to the manuscript, including a new Materials and Methods section (Lines 69 -152)

Table 1B shows data for three viral genes. These data are not explained or discussed in the manuscript. Why are these values in the table and what is their importance?

Response: Thank you for highlighting this issue and our oversight. We believe it is important to show the Ct values of viral genes amplification, because they inversely correlate with viral titer and, therefore, viral load in the infected individuals. Our data underline that individuals with higher levels of IgG tend to show positivity to SARS-CoV-2 with lower viral load. We added two sentences concerning these results: Lines 186-188 and Lines 211-213.

Line 71, sentence starting with "While age,.." is not clear. Specifically, what is "While age" referring to. Please revise.

Response: We revised the sentence of Line 71 in the original manuscript. It now reads: “Adjusting for age and symptoms, having a role as healthcare assistant in our Institute, specifically being a nurse or a physician vs. other professionals (researchers, technicians, administratives), was found to be highly correlated with increased probability of infection (S4 Table, P<0.0001).” (Lines 157-160)

Reviewer #1 

The study provides an interesting picture of the patients admitted to a single centre in 27 months. The study design is appropriated and the results substantially confirm the data of other studies concerning the vaccination against Sars-CoV-2. There are issues to be addressed:

• Concerning the sampling for the diagnostic analyses, how many enrolled subjects underwent naso-oropharyngeal swab and how many were sampled for saliva? If swab and saliva were obtained from the same subjects, did authors observed discrepancies related to the RT-PCR performed for samples correspondent to late stages of infection? Authors should clearly explain this information.

Response: In order to validate the use of saliva samples with our molecular assay for SARS-CoV-2 detection, we performed 2 preliminary tests: i) we analyzed 9 saliva samples collected from symptomatic COVID19 patients, positive for nasopharyngeal swab. All saliva samples (9/9, 100%) confirmed the positivity for SARS-CoV-2; ii) we collected at the same time and analyzed in parallel 47 saliva samples and nasopharyngeal swabs from individuals participating in our study. We obtained concordant results for 96% (45/47) of samples. In details, 100% (34/34) of negative individuals by nasopharyngeal swabs resulted negative also by saliva testing. For the 13 positive individuals by nasopharyngeal swabs: 11 tested positive also by saliva testing with highly comparable results, in terms of Ct amplification, and only 2 samples gave discordant results. The nasopharyngeal swabs of these 2 samples scored positive only for the N viral gene and with amplification at Ct >37, therefore, very close to the limit of detection of our assay. 

Our data agree with published results which show that saliva samples can be successfully employed for SARS-CoV-2 detection by molecular assays with similar or higher sensitivity compared to the same assays applied on nasopharyngeal swabs (Williams E et al., J. Clin. Microbiol. 2020, 58(8): e00776-20; Pasomsub E et al., Clin. Microb Infect 2021, 27 (2): 285.e1-285.e4; Azzi L et al, J Infec 2020, 81(1):e45-e5; Wyllie AL et al, N Engl J Med 2020; 383: 1283-1286). 

Based on these data, we consider nasopharyngeal swab and saliva equivalent for our purposes and discuss them indistinctly throughout our manuscript. However, all cases of possible re-infections (S2 Table) were tested by nasopharyngeal swab. The 30 cases positive post-vaccination (Table 1B), instead, were all tested on saliva except 6 (ID11, ID12, ID14, ID20, ID23, ID30) tested on nasopharyngeal swab. 5 of these cases scored positive for 3 viral genes with Ct<33, except ID23 positive for only 1 gene with Ct>37. Therefore, we believe that our data are comparable independently from the tested specimen. 

We added a comment addressing this issue in the Study Design section (Lines 47-59).

• During the study period different viral clades, including some VOCs such as the alpha variant, emerged and spread. Therefore, did authors observe differences in terms of viral load and serological measurements, among infected subjects in 2020 and those infected in the first part of 2021 (namely, those more probably exposed and infected with alpha variant)?

Response: As commented by the reviewer, in Italy the alpha variant started to spread at the beginning of 2021 and, based on the data released by the Italian National Institute of Health (ISS) reached an incidence of 54% by February 18th 2021 and became prevalent (87%) by March 18th 2021. 

Concerning viral load, from the beginning of our study in April 2020 to the middle of August 2020, all SARS-CoV-2 positive samples showed amplification with Ct >30 for all viral genes. In contrast, starting from August 18th 2020, we observed positive cases with Ct<25 and in some cases Ct<20, suggesting increased viral load. 

From January 7th 2021, the vaccination campaign started in our Institute and, expectedly, IgG levels significantly increased in response to vaccination. These levels were no longer comparable to the levels observed in response to natural viral infections. 

However, in our cohort, before vaccination, the viral titer detected by our molecular tests did not seem to correlate with the intensity of the humoral response, as shown in the Figure below. Panel A shows that in the period between August and December 2020, we, indeed, observed a significant decrease of Ct values for detection of the viral genes compared to the previous months of 2020. However, the levels of IgG in the same periods were very similar (Panel B). Panel C directly compares IgG levels to the average Ct levels of the 3 viral genes in positive individuals in the period between August and December 2020. Again, no correlation is observed between the 2 parameters. Considering these results, we did not address this issue in our manuscript. We are ready to add a comment if required.

• Concerning the possible reinfections, there are three subjects who may be reinfected after less than 100 days. Authors could better discuss if the data reported in table S2 could be more indicative of viral remnants of the first infection. Could these values be non-specific? Instead, concerning those patients putatively reinfected after more days, may these patients be re-infected with a different Sars-CoV-2 variant? Authors should better discuss this section.

Response: The subjects of our study are exclusively healthy adult workers with a median of 41 years of age. We considered a window of >60-days between two SARS-CoV-2 positive samples as a possible reinfection based on previous reports (Lumley SF et al, N Engl J Med 2021; 384: 533-540; Cento V et al. J Infect 2020 Sep;81(3):e90-e92; Chu MC et al. Eur Respir J 2005; Eyre DW, Elife 2020;9: e60675). Although we cannot formally exclude viral remnants of the first infection or false positive amplification, we believe that these are very unlikely events. These details are in the Procedures section of the manuscript (Lines 102-105).

Moreover, unfortunately, we were not able to sequence the virus of the first infection and of the putative re-infection for lack of viral RNA of sufficient amount and quality for preparation of sequencing libraries. Indeed, based on Ct amplification levels, the viral load in these samples was very low (S2 Table). Therefore, we are not able to formally prove two independent infection events or address the interesting issue raised by the reviewer, if these patients have been re-infected with different Sars-CoV-2 variants. However, all re-infections occurred within the very beginning of January 2021, before the large diffusion of the alfa variant of SARS-CoV-2 in Italy. 

• The Discussion section could be improved. Indeed, authors should compare their findings concerning the vaccination with other data reported in literature (e.g. PMID: 34442817, 2021; PMID: 34249017, 2021). For instance, authors should mention the importance of cell-mediated immunity mediators as markers of effective response to vaccination with respect of antibody levels (e.g. PMID: 33534923, 2021). Moreover, authors could discuss if the stratification of the enrolled positive patients according to the viral strain could add major information.

Response: Thanks to the reviewer for this suggestion. We expanded and included the discussion of these issues in our manuscript (Lines 222-230, Lines 244-251, Lines 254-259).

---

## [Editor Report · Decision Letter 1]

11 Jan 2022

Lower probability and shorter duration of infections after Covid-19 vaccine correlate with anti-SARS-CoV-2 circulating IgGs

PONE-D-21-31133R1

Dear Dr. Ronchini,

We’re pleased to inform you that your manuscript has been judged scientifically suitable for publication and will be formally accepted for publication once it meets all outstanding technical requirements.

Kind regards,

Luisa Gregori, Ph.D

Academic Editor

PLOS ONE
---

## [Editor Report · Acceptance letter]

21 Jan 2022

PONE-D-21-31133R1 

Lower probability and shorter duration of infections after COVID-19 vaccine correlate with anti-SARS-CoV-2 circulating IgGs 

Dear Dr. Ronchini:

I'm pleased to inform you that your manuscript has been deemed suitable for publication in PLOS ONE. Congratulations! Your manuscript is now with our production department. 

Kind regards, 

on behalf of

Dr Luisa Gregori 

Academic Editor

PLOS ONE